# Influence of Sourdough Fermentation-Induced Dephytinization on Iron Absorption from Whole Grain Rye Bread–Double-Isotope Crossover and Single-Blind Absorption Studies

**DOI:** 10.3390/nu17243891

**Published:** 2025-12-12

**Authors:** Michael Hoppe, Ann-Sofie Sandberg, Lena Hulthén

**Affiliations:** 1Department of Gastroenterology and Hepatology, Unit of Clinical Nutrition, Sahlgrenska University Hospital, 413 45 Gothenburg, Sweden; 2Department of Internal Medicine and Clinical Nutrition, Sahlgrenska Academy, University of Gothenburg, 405 30 Gothenburg, Sweden; lena@hulthen.st; 3Department of Life Sciences, Food Science, Chalmers University of Technology, 412 96 Gothenburg, Sweden

**Keywords:** non-heme iron, iron status, iron absorption, iron isotopes, whole grain, dietary intervention, women, dephytinization, phytic acid

## Abstract

**Background/Objectives**: There are substantial beneficial health effects from a diet rich in whole grains. However, a high intake of whole grain, and hence a high intake of the iron absorption inhibitor phytate, may result in the impaired bioavailability of non-heme iron. The study examined non-heme iron absorption in healthy women from two portions (80 g and 120 g) of identical whole grain bread, baked with or without phytate-degrading techniques. **Methods**: The study included two single-blinded iron isotope trials. Subjects were served meals containing whole grain rye bread, which was either baked from scalded flour or sourdough-fermented flour labeled with ^55^Fe or ^59^Fe. The absorption of non-heme iron from the meals was measured through the erythrocyte incorporation of radioiron isotopes. **Results**: Iron absorption from the 80 g high-phytate bread was 7.0 ± 4.1% (mean ± SD, *n* = 8). Iron absorption from the 80 g dephytinized bread was 19.1 ± 15.1% (mean ± SD) and thus on average 2.8 times higher compared to the absorption from the high-phytate bread (*p* = 0.001). Iron absorption from the 120 g high-phytate bread was 4.6 ± 2.9% (mean ± SD, *n* = 17). Iron absorption from the 120 g dephytinized bread was 15.0 ± 9.2% (mean ± SD) and thus on average 3.5 times higher compared to the absorption from the high-phytate bread (*p* = 0.001). **Conclusions**: Iron uptake was significantly higher from dephytinized bread compared to scalded bread. And the higher the amount of phytate, the higher the beneficial effects on iron absorption from dephytinization.

## 1. Introduction

To help prevent certain lifestyle-related diseases [1,2], current recommendations advise consuming 25–38 g of dietary fiber per day and ensuring that at least half of all grains consumed are whole grains [3,4,5,6]. According to the Nordic Nutrition Recommendations (2023), the suggested minimum intake of whole grains is 90 g per day [3]. However, research also indicates that a high intake of dietary fiber and whole grains and consequently a high intake of phytate, the major inhibitor of iron absorption, may impair iron status, particularly among individuals with elevated iron requirements [7,8,9,10]. As early as the 1940s, Widdowson and McCance proposed that phytate was responsible for the inhibitory effect of bran on iron absorption in humans [11]. This mechanism is explained by the chelating properties of phytic acid, which binds iron (and other minerals) through its negatively charged phosphate groups, forming stable, insoluble complexes that hinder intestinal iron absorption [12,13].

A wide range of meals, foods, and dishes has been examined regarding the effect of phytate on iron absorption [14,15,16]. Several food processing techniques have proven effective in degrading phytate in grain products [17,18,19,20]. In human studies on phytic acid and iron absorption from bread, the common approach has been to add phytate to different types of bread. For example, Hallberg et al. demonstrated that iron absorption was markedly inhibited when phytate was added in varying amounts to low-extraction white wheat flour bread [15]. Furthermore, comparisons of iron absorption from breads baked with different flours and fermentation methods against a control bread made from low-extraction wheat flour have shown that iron absorption increases as phytate content decreases [16].

To our knowledge, no published data exist on single-meal iron absorption from the same whole grain bread baked with or without phytate-degrading techniques. Therefore, in a single-blinded iron isotope trial (Trial 1 and Trial 2), we investigated iron absorption from two high-extraction whole grain breads, A and B, prepared with identical ingredients. Bread A was baked using scalded flour, while bread B was sourdough fermented. In Trial 1, participants consumed 80 g of bread A and B, and, in Trial 2, 120 g of bread A and B was served.

## 2. Materials and Methods

### 2.1. General Protocol

This study consisted of two single-blinded, crossover trials using the double-radioiron-isotope methodology in young, healthy Swedish women. This design allowed each participant to serve as her own control. The aim was to examine non-heme iron absorption from two whole grain rye breads prepared with ingredients: one dephytinized bread and one bread made from scalded rye. Dephytinization was achieved through sourdough fermentation, which completely degraded phytates, whereas scalding (pouring boiling water over flour before mixing the dough) was used in the other bread to inactivate phytase.

In each trial, participants consumed a standardized light meal on four consecutive mornings. The meal consisted of whole grain rye bread served with Flora^®^ table margarine (15 g), orange marmalade (20 g), and a glass of water (200 mL). Two breads were studied: whole grain bread made from scalded rye (A) and dephytinized whole grain bread (B). Each bread was served on two mornings, following one of the sequences ABBA, BAAB, AABB, or BBAA. The mean daily iron absorption from the two days for each bread type was calculated to minimize the known day-to-day variation in iron absorption [21,22].

To quantify iron absorption, the bread meals were homogeneously labeled with extrinsic radioisotypes, ^59^Fe (for bread B) and ^55^Fe (for bread A), applied as FeCl_3_ by pipetting immediately before serving. Each meal was consumed after an overnight fast (no food after 22:00 and no fluids after 24:00 the previous evening). No food or drink was permitted for three hours following the meal.

In Trial 1, participants consumed 80 g of whole grain rye bread per meal. The intrinsic iron in the dephytinized bread was labeled with ^55^Fe, and the iron in the bread made from scalded rye was labeled with ^59^Fe. In Trial 2, the portion size was increased to 120 g of bread per meal, with the same isotope labeling protocol.

Blood samples were collected approximately 14 days after meal administration, totaling 120 mL per participant. This timing allows for the maximal incorporation of absorbed isotopes into circulating erythrocytes. Brise and Hallberg demonstrated that, by 10–14 days, most absorbed iron is incorporated into red blood cells, providing a reliable estimate of absorption [23]. Sampling earlier would underestimate absorption because a significant fraction of iron would still reside in plasma or bone marrow. Thus, this interval reflects the time required for absorbed iron to be utilized in hemoglobin synthesis and for newly formed erythrocytes to enter the circulation. For an illustration of the study design, see Figure 1.

### 2.2. Inclusion/Exclusion Criteria

Participants were required to be healthy, non-pregnant, non-lactating women of reproductive age without any metabolic, gastrointestinal, or malabsorptive disorders. Individuals were excluded if they had donated blood within two months prior to the study or if they were taking any medications or dietary supplements (including iron) during the study or within two weeks before enrollment. Additional exclusion criteria included infection or inflammation, due to their pronounced effects on iron metabolism [24,25]. To minimize systematic error from infections, acute-phase biomarkers were assessed (C-reactive protein [CRP] and alpha 1-acid glycoprotein [AGP]). Furthermore, prior to serving the bread meals, participants completed a health questionnaire to report any symptoms of infection (e.g., common cold or fever) in the preceding weeks.

### 2.3. Iron Absorption Measurement

This study employed the double-isotope technique, a well-validated method used in human research for over 50 years [21,26]. Iron absorption from the ^59^Fe-labeled bread meal (B) was calculated as the percentage of detected radioactivity in blood samples collected from participants. The radiation from ^59^Fe corresponds to the amount of absorbed ^59^Fe, which, according to the “pool concept”, reflects the total iron absorbed from the bread meal [27,28]. A wet chemical analysis of ^55^Fe and ^59^Fe was performed using a modified version of the method described by Eakins et al. [29]. Duplicate whole blood samples corresponding to 10 mg of iron were transferred to Kjeldahl flasks. Ten milliliters of concentrated H_2_SO_4_ and 18 mL of concentrated HClO_4_ were added, and the mixture was left overnight before digestion using the Tecator system. After cooling, 5 mL of distilled water was added, and the contents were transferred to centrifuge tubes placed in elongated plexiglass racks. Each flask was rinsed three times with 10 mL of distilled water. Bromocresol purple was added to the centrifuge tube, followed by titration with 10 M NH_4_OH until a color change to violet occurred. Approximately 5 mL of NH_4_OH was then added in excess to precipitate iron as Fe(OH)_3_. The tubes were left overnight, then centrifuged for 30 min at maximum speed. The supernatant was carefully removed, and the precipitate was washed three times with 10 mL distilled water. After standing overnight, centrifugation was repeated, and the water was removed. Subsequently, 0.5 mL of concentrated H_3_PO_4_ was added to dissolve the precipitate, and the solution was transferred to Packard vials. Additional rinses with H_3_PO_4_ and NH_4_Cl in ethanol were performed, and the contents were mixed thoroughly. Scintillation gel was added to each vial before counting in the Packard system, ensuring uniform suspension. Vials were then analyzed using a liquid scintillation counter (tri-carb model 1900 TR Packard instruments, Woonsocket, RI 02895, USA) to determine the radiation from ^55^Fe and ^59^Fe. The percentage of iron absorbed was calculated for each participant based on estimated blood volume, determined from height, weight, and hemoglobin concentration [30].

### 2.4. Total Administered Radioactivity

The total administered radioactivity for each participant was 2 μCi from ^55^Fe and 2.0 μCi from ^59^Fe.

### 2.5. The Bread

Bread meal A contained the natural amount of phytic acid (38 mg phytate phosphorus per 100 g bread, equivalent to 138.5 mg phytic acid). Bread meal B consisted of sourdough-fermented bread, which contained 0 mg phytate phosphorus per 100 g bread. The composition of the breads administered in Trial 1 and 2 is presented in Table 1.

The bread was prepared using wholegrain rye flour, wholegrain wheat flour, wheat flour, water, yeast, salt, liquid margarine, and yellow syrup. Bread A was produced from rye scalded at 100 °C to deactivate endogenous phytase, whereas bread B was sourdough-fermented by incorporating 10% sourdough and fermenting the dough for 24 h. Both doughs were fermented with fresh commercial baker’s yeast (Yeast Company, Rotebro Sweden) at 37 °C for 50 min. The baking of breads A and B was performed at 200 °C for 55 min. All flour was sourced from the Lantmännen Cerealia group (Stockholm, Sweden) under the Kungsörnen brand. Phytate (inositol hexa phosphates) and its degradation products were quantified using a method developed at the Department of Life Sciences, Food Science, Chalmers University of Technology, Gothenburg, Sweden, employing high-performance ion chromatography (HPIC) [31]. Samples (0.5 g of freeze dried, ground bread) were pretreated through an extraction with 0.5M HCl followed by centrifugal ultrafiltration and analyzed using HPIC with gradient elution. The inositol phosphates were detected using ultraviolet detection after postcolumn reaction. Mineral content was analyzed using HPIC coupled with UV-vis detection. Organic acids were determined with HPLC according to Scheers et al. [32]. Dietary fiber content was analyzed following the method of Theander et al. (1995) [33]. The breads were baked by a commercial bakery (Lantmännen Cerealia, Stockholm, Sweden) at their modern product development facility in Malmö, Sweden, and subsequently transported and stored in freezers (−20 °C) in Gothenburg until use.

### 2.6. Anthropometric and Laboratory Measurements

Body weight and height were measured, with participants wearing light clothing and no shoes. The following blood parameters were analyzed: hemoglobin concentration (Hb), total iron binding capacity (TIBC), transferrin saturation (TSAT), serum iron concentration (S-Fe), soluble transferrin receptor (sTfR), and serum ferritin concentration (SF). To rule out infection or inflammation, acute-phase proteins C-reactive protein (CRP) and alpha-1-acid glycoprotein (AGP) were also measured.

All analyses were performed at an accredited reference laboratory (Clinical Chemistry Laboratory, Sahlgrenska University Hospital, Gothenburg, Sweden) in compliance with ISO/IEC 15189 [34] standards for medical laboratories. At the time of blood sampling, participants were asked about recent infections, such as colds, coughs, sore throats, or fever within the preceding weeks. Individuals reporting infections and/or presenting CRP levels > 5 mg/L or AGP levels > 1.2 g/L were excluded. See Table 2.

### 2.7. Statistics

The normality of data distribution was assessed using the Shapiro–Wilk test. Differences in iron absorption were analyzed with paired Student’s *t*-tests at a 95% confidence level. Comparisons between study groups were performed using independent Student’s *t*-tests, also at a 95% confidence level. Iron absorption variables were significantly skewed. Consequently, data were log-transformed before statistical analysis. For ease of interpretation, untransformed data are presented as means, standard deviations (SDs), and standard errors of the mean (SEMs). All *p*-values were two-tailed, and statistical significance was defined as *p* < 0.05. Statistical analyses were performed using IBM1 SPSS1 Statistics for Windows 29.0.0 (SPSS Inc., Chicago, IL, USA).

## 3. Results

The results are shown in Table 1. Iron absorption is presented as the non-heme iron absorption from each of the two bread types, as well as the ratio of sourdough bread (bread B) to scalded bread (bread A) absorption.

### 3.1. Trial 1

In Trial 1, nine women were recruited; one was excluded for failing to comply with the fasting requirement on one of the test mornings. Accordingly, the results for Trial 1 are based on eight women aged 20–35 years, with hemoglobin values ranging from 112 to 139 g/L and a serum ferritin concentration between 7 and 55 μg/L. The mean fractional absorption of non-heme iron from 80 g high-phytate bread was 7.0 ± 4.1% (*n* = 8, mean ± SD). In contrast, the absorption from the same amount of dephytinized, low-phytate bread was 19.1 ± 15.1% (mean ± SD), representing an average increase of 2.8-fold compared to the high-phytate bread (ratio = 2.84, *p* < 0.001). See Table 3.

### 3.2. Trial 2

In Trial 2, a total of 21 women were initially recruited. The absorption values for four participants were unavailable due to low radioactivity counts. Therefore, the results for Trial 2 are based on 17 women aged 19–28 years, with hemoglobin values ranging from 120 to 151 g/L and serum ferritin concentrations between 9 and 100 μg/L. The mean fractional absorption of non-heme iron from 120 g high-phytate bread was 4.6 ± 2.9% (mean ± SD, *n* = 17). In contrast, the absorption from the dephytinized bread was 15.0 ± 9.2% (mean ± SD), representing an average increase of 3.5-fold compared to the high-phytate bread (ratio = 3.47, *p* < 0.001). See Table 4.

The sourdough-to-scalded iron absorption ratio was significantly higher in Trial 2 (ratio = 3.47, *p* = 0.020) compared to Trial 1, which used 80 g bread (ratio = 2.84).

## 4. Discussion

This study demonstrates that sourdough fermentation, which effectively removes phytate from whole grain rye bread, significantly enhances fractional non-heme iron absorption compared to scalded rye bread. These findings confirm the strong inhibitory effect of phytate on iron absorption and underscore the potential of sourdough fermentation as a practical strategy to improve iron bioavailability in high-phytate cereal products.

### 4.1. Effect of Phytate Reduction

The improvement in iron absorption was most pronounced when comparing breads with a near-complete phytate removal (<1 mg phytate-P, equivalent to <3.55 mg phytic acid) to breads with a high phytate content (47 mg phytate-P, equivalent to 167 mg phytic acid). The absorption ratio in this comparison was 3.47, whereas the ratio was 2.84 when comparing sourdough bread to bread containing 31 mg phytate-P (110 mg phytic acid). These results align with previous observations that the first 10–20 mg of phytate-P in a meal exerts the strongest inhibitory effect, while additional phytate has a diminishing impact. Above 40–50 mg phytate-P, further inhibition is modest [15,35].

### 4.2. Comparison with Algorithm Predictions

Using the iron absorption algorithm developed by Hallberg et al. [35] (inputting the content of iron, phytate, ascorbic acid, and calcium in the bread), the calculated absorption from 80 g of scalded bread is 8.1%, while the calculated iron absorption from 80 g of sourdough bread is 22.4%, yielding an absorption ratio of 2.77 (this is very similar to the present study’s result, which showed a ratio of 3.47 for the larger portion size). Thus, the calculated absorption values using Hallberg’s algorithm closely match the observed results, supporting the validity of these findings. The minor discrepancies likely reflect the complexity of high-extraction rye bread, which contains bran and other phytochemicals that may influence absorption. This suggests that, while algorithms provide useful estimates, real-world food matrices introduce variability.

### 4.3. Phytic Acid-to-Iron Molar Ratio

Siegenberg et al. reported a two-fold increase in iron absorption from a white bread meal when the phytic acid-to-iron molar ratio was reduced from 3.1 to 0.5 (phytate phosphorus decreased from 70 mg/100 g to 11 mg/100 g, equivalent to 249 mg and 39 mg phytic acid) [36]. Larsson et al. found no effect when lowering the ratio from 9.5 to 3.1 in oat porridge, but reducing it to 1.7 increased absorption by 36% [37]. Hurrell concluded that near-zero phytate is ideal for iron bioavailability; if not possible, the ratio should be below 1.0, and preferably under 0.4 [38].

In this study, the bread iron content was low (1.2 mg/100 g), resulting in a high ratio in scalded bread. Sourdough bread, however, achieved a phytic acid-to-iron molar ratio below 1.0, which is considered optimal for iron bioavailability. In contrast, scalded bread had ratios of 9–11, strongly inhibiting absorption. Our findings confirm that near-complete phytate removal through sourdough fermentation can achieve these favorable ratios.

### 4.4. Impact of Portion Size

Increasing the portion size from 80 g to 120 g doubled the phytate content in scalded bread, thereby increasing its inhibitory effect. However, the phytic acid-to-iron molar ratio did not double as iron content increased proportionally. This illustrates that both the absolute phytate content and its ratio to iron influence absorption. The larger portion size resulted in a greater difference in the absorption ratio between sourdough and scalded bread, but the practical significance depends on the total iron absorbed rather than relative differences.

### 4.5. Absolute Iron Absorption and Nutritional Relevance

Although fractional absorption was substantially higher from sourdough bread, the absolute amount of iron absorbed per meal remained small due to the low iron content of rye bread (approximately 1.2 mg/100 g). For example, 80 g of sourdough bread provided roughly 0.21 mg of absorbed iron (22.4% of 0.96 mg), while 80 g of scalded bread provided only 0.08 mg (8.1% of 0.96 mg). Even with a two- to three-fold increase in absorption, these amounts are insufficient to significantly improve iron status over time. This could explain why our previous 12-week intervention using the same breads showed no improvement in iron status [39]. The discrepancy between single-meal absorption studies and long-term interventions underscores that fold increases in absorption, while scientifically interesting, have limited practical relevance unless the total absorbed iron is substantial.

### 4.6. Other Potential Influences

Sourdough fermentation also increased lactic acid content, which has been proposed to promote iron absorption. However, the current evidence from both in vitro and human studies does not support a significant role for lactic acid in enhancing iron absorption [36,40]. Additionally, the use of high-extraction rye flour introduces potential confounding factors from bran components and phytochemicals, which may influence iron absorption independently of phytate content. As noted by Brune et al. [16], this potential confounding effect could be minimized by using bread made from low-extraction flour, which primarily contains the innermost part of the starchy endosperm. Such an approach would reduce interference from other dietary factors and improve the accuracy of measuring the isolated inhibitory effect of phytates. On the other hand, because low-extraction flour has a relatively low phytate content, a further reduction through sourdough fermentation would not have been possible. Unlike studies that add varying amounts of phytate to bread or that compare high-extraction high-phytate bread with a low-extraction control, our objective was to investigate the effect of sourdough fermentation versus scalding using identical ingredients.

### 4.7. Comparison with Previous Studies

Our findings are consistent with earlier research demonstrating that reducing the phytate content or lowering the phytic acid-to-iron molar ratio improves iron absorption [36,37]. Siegenberg et al. [36] reported a two-fold increase in absorption when reducing the ratio from 3.1 to 0.5, and Hurrell concluded that a near-zero phytate content or a ratio below 1.0 is desirable [38]. The present study confirms these principles in a real food matrix using sourdough fermentation as a practical approach.

### 4.8. Strengths and Limitations

A key strength of this study is the use of a controlled crossover design with radioiron isotopes, enabling precise measurements of fractional absorption. However, the complexity of high-extraction rye bread may introduce confounding factors, and its low iron content limits the nutritional impact of the observed differences. Furthermore, single-meal studies may exaggerate the effects of dietary inhibitors compared to habitual diets [41].

### 4.9. Implications for Dietary Strategies

Although sourdough fermentation markedly improves fractional iron absorption from high-phytate rye bread, its practical impact on iron status is constrained by the low iron content of bread and the small absolute amount absorbed per meal. Strategies to improve iron status should therefore combine phytate reduction with increased iron content or the inclusion of enhancers such as ascorbic acid. Alternatively, sourdough fermentation could be applied to other cereal-based foods with a higher iron content to maximize its benefit.

## 5. Conclusions

Sourdough fermentation effectively removes phytate from whole grain rye bread and significantly enhances non-heme iron absorption. Eliminating phytate through sourdough fermentation improves iron bioavailability and thereby increases the nutritional value of high-extraction whole grain rye bread beyond its already well-established health benefits. However, the nutritional relevance of this improvement is limited by the low iron content of bread. Future research should focus on combining phytate reduction with strategies that increase the total iron intake to achieve meaningful improvements in iron status.

## Figures and Tables

**Figure 1 nutrients-17-03891-f001:**
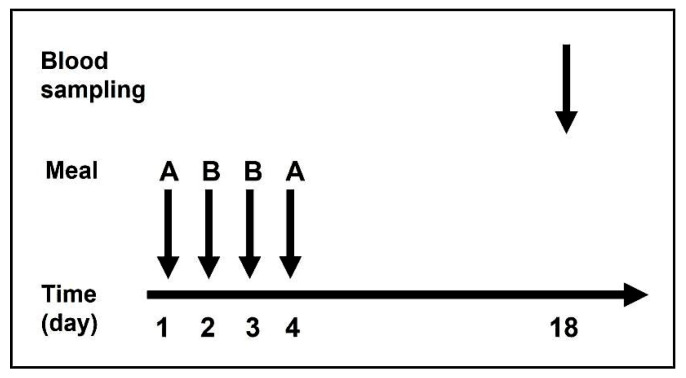
Study design.

**Table 1 nutrients-17-03891-t001:** Bread composition.

	Trial 1 (n = 8)		Trial 2 (n = 17)	
	Meal A	Meal B	Meal A	Meal B
Administered amount of bread/meal (g)	80	80	120	120
Whole grain (g)	30	30	44	44
Phytate phosphorus (mg) ***	31	<1.0	47	<1.0
Lactic acid (mg)	5	920	7	1380
Non-heme Fe (mg)	1.0	1.0	1.4	1.4
Ascorbic acid (mg)	1.0	1.0	1.6	1.6
Calcium (mg)	4	4	6	6
Molar ratio phytic acid */Fe	9.3:1	<0.1:1	11.1:1	<0.1:1

* 1 mg phytate phosphorus = 3.55 mg phytic acid. The scalded bread in meal A contained 1.93 µmol InsP6/g and 0.15 µmol InsP5/g (mean value of five bread samples), corresponding to 0.36 mg and 0.02 mg phytate-P/g, respectively. The sourdough bread meal B contained no detectable amounts of inositol phosphates.

**Table 2 nutrients-17-03891-t002:** Subject characteristics *.

	Trial 1 (*n* = 8)	Trial 2 (*n* = 17)
Age (years)	24.4 ± 4.9 (20–35)	23.6 ± 2.4 (19–28)
Height (cm)	170.8 ± 3.5 (166–176)	168.4 ± 6.0 (161–180)
Weight (kg)	60.8 ± 5.7 (49–66)	61.6 ± 9.3 (45–84)
Hb (g/L)	129.5 ± 8.7 (112–139)	131.5 ± 7.1 (120–151)
TSAT (%)	21.8 ± 16.9 (5–58)	11.2 ± 13.1 (8–43)
SF (ug/L)	30.4 ± 17.4 (7–55)	40.5 ± 27.1 (9–100)
S-Fe (µmol/L)	15.1 ± 8.8 (4–31)	18.6 ± 9.3 (6–44)
TIBC (µmol/L)	77.6 ± 16.2 (54–110)	77.6 ± 11.7 (66–110)
sTfR (mg/L)	4.0 ± 3.2 (2–12)	3.1 ± 0.7 (2–5)
CRP (mg/L)	1.5 ± 0.7 (1–2)	2.7 ± 1.1 (2–4)
AGP (g/L)	0.6 ± 0.1 (0.4–0.8)	0.7 ± 0.1 (0.4–1.0)

* Mean ± SD and range (Min-Max). Hemoglobin concentration (Hb); transferrin saturation (TSAT); serum ferritin concentration (SF); serum iron concentration (S-Fe); total iron binding capacity (TIBC); soluble transferrin receptor (sTfR); C-reactive protein (CRP); alpha 1-acid glycoprotein (AGP).

**Table 3 nutrients-17-03891-t003:** Trial 1—Effect of dephytinization on iron absorption from 80 g whole meal rye bread.

Subject	Iron Absorption (%)		Iron Absorption Ratio	*p*-Value
	Meal A	Meal B	(Meal B/Meal A)	
1	9.0	23.3	2.59	
2	5.6	10.5	1.88	
3	5.5	9.8	1.78	
4	5.1	10.8	2.12	
5	1.1	5.9	5.36	
6	15.1	52.8	3.50	
7	8.6	23.6	2.74	
8	5.8	15.8	2.72	
Mean	7.0	19.1	2.84	<0.001
SD	4.1	15.1	1.16	
SEM	1.4	5.3	0.41	

**Table 4 nutrients-17-03891-t004:** Trial 2—Effect of dephytinization on iron absorption from 120 g whole meal rye bread.

Subject	Iron Absorption (%)		Iron Absorption Ratio	*p*-Value
	Meal A	Meal B	(Meal B/Meal A)	
1	0.7	2.8	4.00	
2	7.5	22.7	3.03	
3	6.1	25.3	4.15	
4	0.5	2.2	4.40	
5	7.2	18.2	2.53	
6	0.6	2.1	3.50	
7	0.6	1.8	3.00	
8	7.8	17.3	2.22	
9	3.2	13.0	4.06	
10	6.6	18.4	2.79	
11	3.3	14.8	4.48	
12	3.3	15.1	4.58	
13	8.1	20.1	2.48	
14	6.1	17.1	2.80	
15	5.1	14.9	2.92	
16	8.8	36.4	4.14	
17	3.3	13.0	3.94	
Mean	4.6	15.0	3.47	<0.001
SD	2.9	9.2	0.79	
SEM	0.7	2.2	0.19	

## Data Availability

The original contributions presented in this study are included in the article. Further inquiries can be directed to the corresponding author.

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
