# Peer review of "Influence of Sourdough Fermentation-Induced Dephytinization on Iron Absorption from Whole Grain Rye Bread–Double-Isotope Crossover and Single-Blind Absorption Studies"

_nutrients, 2025, doi:10.3390/nu17243891_

Round 1

Reviewer 1 Report

Comments and Suggestions for Authors

This is an interesting study that provides useful information on the positive effects on iron absorption through the removal of phytate by sourdough fermentation. The methodology is sound and the results are clear but the manuscript could be improved by taking into account the following comments:

Major points:

  1. The analysis of phytate in the bread should be given in more detail i.e. the content of different IPs (as it is only IP6, 5 and possibly 4 that inhibit iron absorption).
  2. The iron absorption results should be given in a separate table/figure. For the first study there were only 8 subjects, so a figure showing the absorption of iron from the two types of bread for each individual could be shown on a graph. Possibly the second study could be presented in the same way. In any event, SDs should be given, not SEMs.
  3. The discussion is too long and unfocused. It would benefit from restructuring and review by a native English speaker. The authors are encouraged to agree on the main findings that they want to draw out of the work, and to discuss them point by point. One important issue is the fact that their previous intervention study showed no effect of low phytate bread on iron status. They need to put this in context by commenting on the quantity of iron absorbed from the breads (as the intake is rather low) instead of focussing on fold increases in absorption. The total amount of iron absorbed is the critical issue.

Minor points:

  1. Line 61, data is a plural word so should have 'are' after it, not 'is'.
  2. Line 64. Add 's' to breads.
  3. Line 65. Many readers will not be familiar with 'scalded' flour. Please describe this early on in the manuscript.
  4. Line 73. Change 'later' to latter.
  5. Line 75. Flora is a trade name so add the appropriate symbol.
  6. Line 96. Assuming the women were not menstruating at the time of the study, please rephrase this description.
  7. Lines 101-104. Needs editing to make clearer.
  8. Line 125. Composition not 'content'.
  9. Table 2. Typo for weight.
Comments on the Quality of English Language

Mostly fine but there are a few sentences that could be improved by review from a native English speaker.

Author Response

We thank the reviewer for very valuable comments and suggestions which we here have taken into consideration.

Comments 1: Line 71-73: “The objective was to examine non-heme iron absorption from dephytinized whole meal rye bread, whole grain bread made from scalded rye. The later thus containing natural high amount of phytic acid.” - the purpose of the study must be clear, and in this form it is not understandable. It should be improved.

- it would be useful to include a schematic representation of the study design in the “General protocol” section

Response 1: In the revised manuscript the sentence on line 71 has been rephrased. Hopefully this is making the objective of the study clearer.

Comments 2: Line 93-94: “In both studies blood samples were drawn approximately two weeks after the bread meal administration. “ - Please explain why two weeks after the meal? What are the indications for this? It's best to support this with relevant literature.

Response 2: The rationale for collecting blood samples 14 days after meal administration is to allow maximal incorporation of absorbed isotopes into circulating erythrocytes. Brise and Hallberg demonstrated that by 10–14 days, most absorbed iron is incorporated into red blood cells, providing a reliable estimate of absorption [ Brise, H. and Hallberg, L. Determinations of Fe55 and Fe59 in blood. Int J Appl Rad Isotopes, 1960. 9:p. 100-108]. Sampling earlier would underestimate absorption because a significant fraction of iron would still be in plasma or marrow. Thus, this interval reflects the time required for absorbed iron to be utilized in hemoglobin synthesis and for newly formed erythrocytes to enter the bloodstream. This is now incorporated (line 98) into the revised manuscript.

Comments 3: “The total amount of blood drawn was 120 mL.” - from each participant?

Response 3: That is correct that the total amount of blood drawn was 120 ml from each participant. The rationale for this amount is firstly that an amount of blood corresponding to 10 mg iron in duplicate is needed for the wet chemical analysis. Depending on Hb-concentration this usually range between 20-30 ml blood, which means that in a subject with low Hb an amount of 60 ml is needed for the analysis. And just as a precaution in case of any unfortunate mishap during the analysis, the double amount is drawn so that there is always the possibility of redoing the analysis.

Comments 4: Line 95: Inclusion/exclusion criteria – please explain the reason for only young women participating in the study

Response 4: We recruited women of fertile age because this group represents the population most vulnerable to iron deficiency and iron-deficiency anemia worldwide. Also, since women of fertile age in general have low iron stores, they absorb a much higher percentage of iron from meals than men or postmenopausal women do. For example, if you want to compare iron absorption from a high phytate meal vs. a low-phytate meal, the differences can be 2- to 5-fold in women with low stores, but almost undetectable in iron-replete men.

Comments 5: Line 113: Whole blood duplicates corresponding to 10 mg of Fe were pre-treated” – please explain the pre-treatment process

Response 5: In the revised manuscript we have added a section describing the different steps in the wet chemical analysis. See line 129-158.

Comments 6: Line 129-140:  Baking details are available in reference 36, but it would be more convenient and faster for readers if they were also briefly presented in this manuscript. What do you mean by "yellow syrup"? Fresh or instant yeast was used in the recipe? Please add the manufacturers of the individual bread ingredients, the equipment used for baking and fermentation, and details of the fermentation conditions.

Response 6: In the revised manuscript we have added a section under “The bread” that hopefully describes the breads and the baking details clearer.

Yellow syrup is a thick, amber-colored, almost yellow, sweetener with a rich, caramel-like flavor. It's made from refined sugar and is used to add moisture, sweetness, and a golden color to baked goods.

Fresh yeast was used in the recipe, and this information is now added in the revised manuscript.

Comments 7: Table 1 - Please provide table title

Response 7: Table title has now been provided.

Comments 8: Table 2 - Were the same participants in trials 1 and 2? Please add ranges in each parameter [in brackets] and explain the differences in CRP and TSAT between trials.

Response 8: It was not the same participants in the two different trials. It was two different groups of women. Ranges for each parameter has now been incorporated. Reason for the differences in CRP and TSAT was due to incorrectly entered values. The correct values ​​are now reported in the revised table

Comments 9: Line 180: “…results in Trial 2 are based on eight females “- do you mean results in Trail 1?

Response 9: The reviewer is correct. It should say table 1 and not 2. This is now corrected.

Comments 10: Line 183 and 192: “dehytinized” – please check 

Response 10: In the revised manuscript this is now corrected.

Comments 11: Line 188: “the results in Trial 1” - rather Trial 2

Response 11: In the revised manuscript this is now corrected.

Comments 12: Line 229-230: “this bread could possibly contain yet unknown iron absorption confounding dietary factors” – any speculation?

Response 12: This sentence is now removed. There are none of the already established iron absorption affecting dietary factors present in the bread. And we find it difficult to speculate on factors not analyzed in the breads.

Reviewer 2 Report

Comments and Suggestions for Authors

Whole-grain flour products offer health benefits, including increased intake of minerals and dietary fibre. Still, their high phytate and fibre content can reduce the absorption of minerals like calcium, iron, and zinc by forming insoluble salts. Phytates, found in plant foods, including cereals and flours, bind to minerals, a process that has led to the concern that excessive whole-grain consumption might hinder nutrient uptake. Despite this, the overall health advantages of whole grains often outweigh the potential drawbacks.

The study raises an important issue: impaired bioavailability of iron from whole grain products. Indeed, a proper absorption of dietary iron is crucial for individuals with high iron requirements (pregnant women, children, and adolescents). The topic of the manuscript (nutrients-4011990) is original, presents the study that examined the absorption of non-heme iron from whole-grain rye bread baked with high-phytate (scalded) rye flour or dephytinized sourdough rye flour in healthy women. In this study,  a validated double-isotope technique was utilised. Bread was labelled with two iron isotopes (55Fe or 59Fe), and the absorption of non-heme iron from meals was measured by radioactive iron incorporation into erythrocytes.

The manuscript is interesting, its structure is coherent and clear, and the language is appropriate and clear, facilitating reading and interpretation of the results. Descriptions of the results are generally readable enough to allow a full understanding and interpretation of the article. However, some aspects of the manuscript need to be improved and clarified:

Line 71-73: “The objective was to examine non-heme iron absorption from dephytinized whole meal rye bread, whole grain bread made from scalded rye. The later thus containing natural high amount of phytic acid.” - the purpose of the study must be clear, and in this form it is not understandable. It should be improved.

- it would be useful to include a schematic representation of the study design in the “General protocol” section

Line 93-94: “In both studies blood samples were drawn approximately two weeks after the bread meal administration. “ - Please explain why two weeks after the meal? What are the indications for this? It's best to support this with relevant literature.

“The total amount of blood drawn was 120 mL.” - from each participant?

Line 95: Inclusion/exclusion criteria – please explain the reason for only young women participating in the study

Line 113: Whole blood duplicates corresponding to 10 mg of Fe were pre-treated” – please explain the pre-treatment process

Line 129-140:  baking details are available in reference 36, but it would be more convenient and faster for readers if they were also briefly presented in this manuscript. What do you mean by "yellow syrup"? Fresh or instant yeast was used in the recipe? Please add the manufacturers of the individual bread ingredients, the equipment used for baking and fermentation, and details of the fermentation conditions.

Table 1 - Please provide table title

Table 2 - Were the same participants in trials 1 and 2? Please add ranges in each parameter [in brackets] and explain the differences in CRP and TSAT between trials.

Results:
Line 180: “…results in Trial 2 are based on eight females “- do you mean results in Trail 1?

Line 183 and 192: “dehytinized” – please check 

Line 188: “the results in Trial 1” - rather Trial 2

Discussion: The discussion is clear and comprehensive, and the drawn conclusions are consistent.

Line 229-230: “this bread could possibly contain yet unknown iron absorption confounding dietary factors” – any speculation?

References are appropriate, but unfortunately, most of the cited sources are older than 10 years.
Self-citations are present but not overused.

Author Response

We thank the reviewer for very valuable comments which we have taken into consideration.

Comments 1: The analysis of phytate in the bread should be given in more detail i.e. the content of different IPs (as it is only IP6, 5 and possibly 4 that inhibit iron absorption).

Response 1: In the revised manuscript we have added a section under “The bread” that hopefully describes this clearer. We have quantified both IP6 and IP5. See Table 1.

Comments 2: The iron absorption results should be given in a separate table/figure. For the first study there were only 8 subjects, so a figure showing the absorption of iron from the two types of bread for each individual could be shown on a graph. Possibly the second study could be presented in the same way. In any event, SDs should be given, not SEMs.

Response 2: In the revised manuscript, we have now given the iron absorption results for Trial 1 and Trial 2 in two separate tables. We have also given SD.

Comments 3: The discussion is too long and unfocused. It would benefit from restructuring and review by a native English speaker. The authors are encouraged to agree on the main findings that they want to draw out of the work, and to discuss them point by point. One important issue is the fact that their previous intervention study showed no effect of low phytate bread on iron status. They need to put this in context by commenting on the quantity of iron absorbed from the breads (as the intake is rather low) instead of focussing on fold increases in absorption. The total amount of iron absorbed is the critical issue.

Response 3: In the revised manuscript, we have restructured the discussion and had the manuscript proofread.

Comments 4: Minor points:

  1. Line 61, data is a plural word so should have 'are' after it, not 'is'.
  2. Line 64. Add 's' to breads.
  3. Line 65. Many readers will not be familiar with 'scalded' flour. Please describe this early on in the manuscript.
  4. Line 73. Change 'later' to latter.
  5. Line 75. Flora is a trade name so add the appropriate symbol.
  6. Line 96. Assuming the women were not menstruating at the time of the study, please rephrase this description.
  7. Lines 101-104. Needs editing to make clearer.
  8. Line 125. Composition not 'content'.
  9. Table 2. Typo for weight.

Response 4: These minor points have now been attended. As well as a short description of scalding which is now given on line 74. And a rephrasing of the description regarding the acut-phase markers (former line 101-104).